# Culturing of a complex gut microbial community in mucin-hydrogel carriers reveals strain- and gene-associated spatial organization

Xiaofan Jin [1,5], Feiqiao B. Yu [2,5], Jia Yan[2], Allison M. Weakley[3], Veronika Dubinkina[1], Xiandong Meng[3] & Katherine S. Pollard [1,2,4] ✉

Microbial community function depends on both taxonomic composition and spatial organization. While composition of the human gut microbiome has been deeply characterized, less is known about the organization of microbes between regions such as lumen and mucosa and the microbial genes regulating this organization. Using a defined 117 strain community for which we generate high-quality genome assemblies, we model mucosa/lumen organization with in vitro cultures incorporating mucin hydrogel carriers as surfaces for bacterial attachment. Metagenomic tracking of carrier cultures reveals increased diversity and strain-specific spatial organization, with distinct strains enriched on carriers versus liquid supernatant, mirroring mucosa/lumen enrichment in vivo. A comprehensive search for microbial genes associated with this spatial organization identifies candidates with known adhesion-related functions, as well as novel links. These findings demonstrate that carrier cultures of defined communities effectively recapitulate fundamental aspects of gut spatial organization, enabling identification of key microbial strains and genes.

Human gut microbiomes consist of diverse microbial taxa[1,2], with typical complexity on the order of one hundred species or more in a single individual[3]. The spatial organization of gut microbes is linked to community function and host health[4–10]. In particular, different taxa are enriched between mucosa and lumen[11–16], with mucosal colonizing bacteria being especially well-positioned to regulate host-microbiome interactions and immunomodulation[17–21]. However, we still lack a high taxonomic-resolution view of ecological differences between lumen and mucosa, and accordingly possess a limited understanding of genetic factors underlying this spatial structure. As the human gut microbiome has dynamic within-species genetic structure[22–25], we hypothesize that distinct spatial organization may (i) occur at the level

of individual strains, and (ii) be associated with specific gene families and pathways that regulate mucosa versus lumen colonization.

To test our hypotheses, we develop an integrated experimental-computational workflow that compares lumen- and mucosal-like niches within a complex gut community. By using metagenomic sequencing, we are able to profile microbes with high taxonomic resolution, enabling strain- and gene-level analysis. We use a synthetic 117-strain community modeled closely after the recently published hCom2 community[26], cultured in vitro with added mucin-agar carriers (i.e., carrier cultures) to provide a mucosal-like substrate for bacterial attachment distinct from the surrounding liquid supernatant[27,28]. To identify genetic correlates of carrier colonization, we implement a

[1]Gladstone Institutes, San Francisco, CA, USA. [2]Chan-Zuckerberg Biohub, San Francisco, CA, USA. [3]Sarafan ChEM-H Institute, Stanford University, Stanford, CA, USA. [4]University of California San Francisco, San Francisco, CA, USA. [5]These authors contributed equally: Xiaofan Jin, Feiqiao B. Yu. ✉e-mail: katherine.pollard@gladstone.ucsf.edu

computational workflow that uses a comprehensive search across KEGG Orthology (KO) gene families[29] to identify associations between gut spatial organization and underlying microbial genotypes, using phylogenetic regression to account for evolutionary relationships between taxa[30–32].

Our approach provides several key advantages: first, by using an in vitro approach that allows mucin carrier and supernatant subpopulations to be independently sampled[28]—analogous to mucosa and lumen in vivo—we obtain information on spatial structure missing from stool sampling and traditional liquid culture. Independent sampling of the lumen and mucosal subpopulations is also possible using in vivo human gut biopsy, but the invasiveness of this approach limits sample sizes and longitudinal measurements[33]. In contrast, our in vitro carrier culture platform enables us to sample mucosal- and lumen-like community subpopulations across multiple passage timepoints, with multiple independent replicates. Second, using our defined 117-strain community with high-quality genomes for each member allows us to emulate the bacterial complexity found in human guts, yet still accurately quantify abundance using metagenomic sequencing even between closely related strains. Strain-level measurements are critical for enabling gene-level analysis, as they allow genetic comparisons between closely related taxa. By comparison, earlier work with carriers used 16S sequencing of undefined communities to produce measurements with more limited taxonomic resolution and did not seek to identify genes associated with carrier colonization[28].

We demonstrate that this approach yields detailed strain-level measurements of differential spatial organization, revealing taxa which are reproducibly enriched or depleted at steady state on mucin carriers relative to supernatant—we refer to this as carrier/supernatant enrichment, or carrier enrichment for short. We compare these results with no-carrier (i.e., liquid media only) cultures and cultures with plain-agar (i.e., no mucin) carriers to disentangle the specific effects of added mucin versus hydrogel surface, revealing that the presence of carriers serves to increase community richness and mucin has some specific effects. Then, we identify numerous genes and biosynthetic gene clusters that distinguish carrier-enriched strains consistently across phylogenetic lineages, including genes related to cell adhesion and biofilm formation whose presence differs between closely related strains with distinct carrier-enrichment profiles. By validating these results in vivo with biopsy data and showing that they are not driven by specific analysis methodologies, we demonstrate the relevance of our flexible in vitro platform to microbial community structure in the human gut.

## Results
### Closed and nearly closed genomes enable strain-level metagenomic profiling of complex-defined microbial communities
Starting from isolate cultures of 123 bacterial strains that are prevalent in the human gut microbiome (Fig. 1a and Supplementary Data S1), we first generate high-quality, contiguous genomes for all strains except 4 that already have closed genomes. For the other 119 strains, we perform hybrid assembly of long Nanopore (median $3.9 \times 10^4$ reads/strain) and short Illumina reads (median $1.7 \times 10^6$ reads/strain) (Supplementary Fig. S1, "Methods"), successfully generating high-quality assemblies with 81 fully closed single contig genomes, 20 2-contig genomes, and overall no genomes with more than 10 contigs. By contrast, the closest available NCBI genome (Fig. 1a) is more fragmented (78/123 comprise more than 10 contigs) and less closely related to the strain in our defined community; 20/123 have >0.1% ANI difference to our strain, and 33/123 contain 100 or more differential KEGG Orthology (KO) gene families (see "Methods" and Supplementary Fig. S2). This reference database of closed and nearly closed genomes that are exact strain matches provides a critical resource for accurate strain and gene-level characterization of metagenomic data in this study and future studies. Next, isolate strains are combined into a single

community using anaerobic automated liquid handling (see "In vitro culture of synthetic community with mucin carriers" and Supplementary Fig. S3) and inoculated into cultures containing 0.5% mucin 1% agar carriers and MEGA media[34] Cultures are serially passaged six times at 3-day intervals. (P1 to P6, see Fig. 1b). As a control, we also culture in parallel the same inoculum with MEGA media only, i.e., liquid-only culture. We use metagenomic sequencing of carriers and supernatant sampled independently ($1.2 \times 10^7$ read pairs per sample) at each passage to quantify strain relative abundances (see "Read mapping and abundance estimation" Supplementary Fig. S4). To analyze read libraries with high taxonomic resolution, we use NinjaMap[26] with our custom genome database to generate strain-level abundances (Fig. 1c and Supplementary Data S2). We observe a high correlation between these results and alternatively using Kraken2[35] with the UHGG database[2] (median $R^2 = 0.987760$ across samples, see Supplementary Fig. S5) or Kraken2 with our custom genome database (median $R^2 = 0.999442$ across samples), validating our choice to use NinjaMap.

### Mucin carriers increase community richness and promote strain co-existence within in vitro cultures
Next, we characterize differences that result from spatial structure introduced by the incorporation of mucin carriers. These experiments use an inoculum with 117 of the 123 strains; 6 strains fail to grow from glycerol stocks as isolates (Supplementary Fig. S7; detection cutoff 0.0001% relative abundance, 1% horizontal coverage). In liquid-media-only cultures without carriers, this richness falls to a median of 55 detected strains by the 4 late passages (passages P3-P6, corresponding to days 9-18). By contrast, carrier cultures seeded with the same inoculum stabilize to a median of 75 and 78 detectable strains on carriers and supernatant, respectively (Fig. 1d). This significantly elevated richness ($P < 1^{-5}$, see Supplementary Fig. S6) approaches that of in vivo results using fecal samples from germ-free mice orally gavaged with hCom2 (median 85/119 detected strains across 19 mice)[26], suggesting cultures with carriers provide a closer analog to in vivo conditions than do liquid-only cultures. Increased richness is particularly noticeable in Firmicutes, Firmicutes_A, and Bacteroidota (see Supplementary Fig. S6), while total abundance is higher for Firmicutes and Firmicutes_A, but lower in Bacteroidota (Fig. 1e).

Beyond phylum-level effects, abundance shifts also occur at the strain level. Addition of carriers increases abundance for a diverse set of strains including *Bacteroides caccae* ATCC-43185, *Lactobacillus ruminis* ATCC-25644, *Coprococcus comes* ATCC-27758 (which displays extremely sticky/slime phenotype in pure culture), two strains in family Marinifilaceae (*Butyricimonas virosa* DSM-23226, *Odoribacter splanchnicus DSM-20712*), and both sulfur reducing bacteria (*Desulfovibrio piger* ATCC-29098 and *Bilophila wadsworthia ATCC-49260* from phylum Desulfobacterota). Some taxa are largely unaffected by carriers, such as three Bifidobacterium strains, while few taxa are negatively affected by carriers, with three closely related Veillonella strains being notable exceptions (see Supplementary Fig. S8). These strain-level abundance shifts do not always align with corresponding phylum-level shifts, emphasizing the value of our highly resolved taxonomic measurements.

One of the most striking abundance shifts revealed by strain-level analysis is the co-existence of closely related strains with the addition of carriers. In liquid-only culture, *Bacteroides dorei DSM-17855* outcompetes two closely related (ANI > 99%) strains, *Bacteroides dorei 5-1-36-D4* and *Bacteroides sp. 9-1-42FAA* (Fig. 1f). By contrast, these three strains coexist stably in culture when carriers are present. Other examples can be found between two closely related (ANI -80%) Firmicutes_A strains: *Subdoligranulum sp. 4-3-54A2FAA* and *Subdoligranulum variabile DSM-15176* (Fig. 1g), and between two closely related (ANI -80%) Firmicutes_C strains: *Acidaminococcus fermentans DSM-20731* and *Acidaminococcus intestini D21* (Fig. 1h). These observations

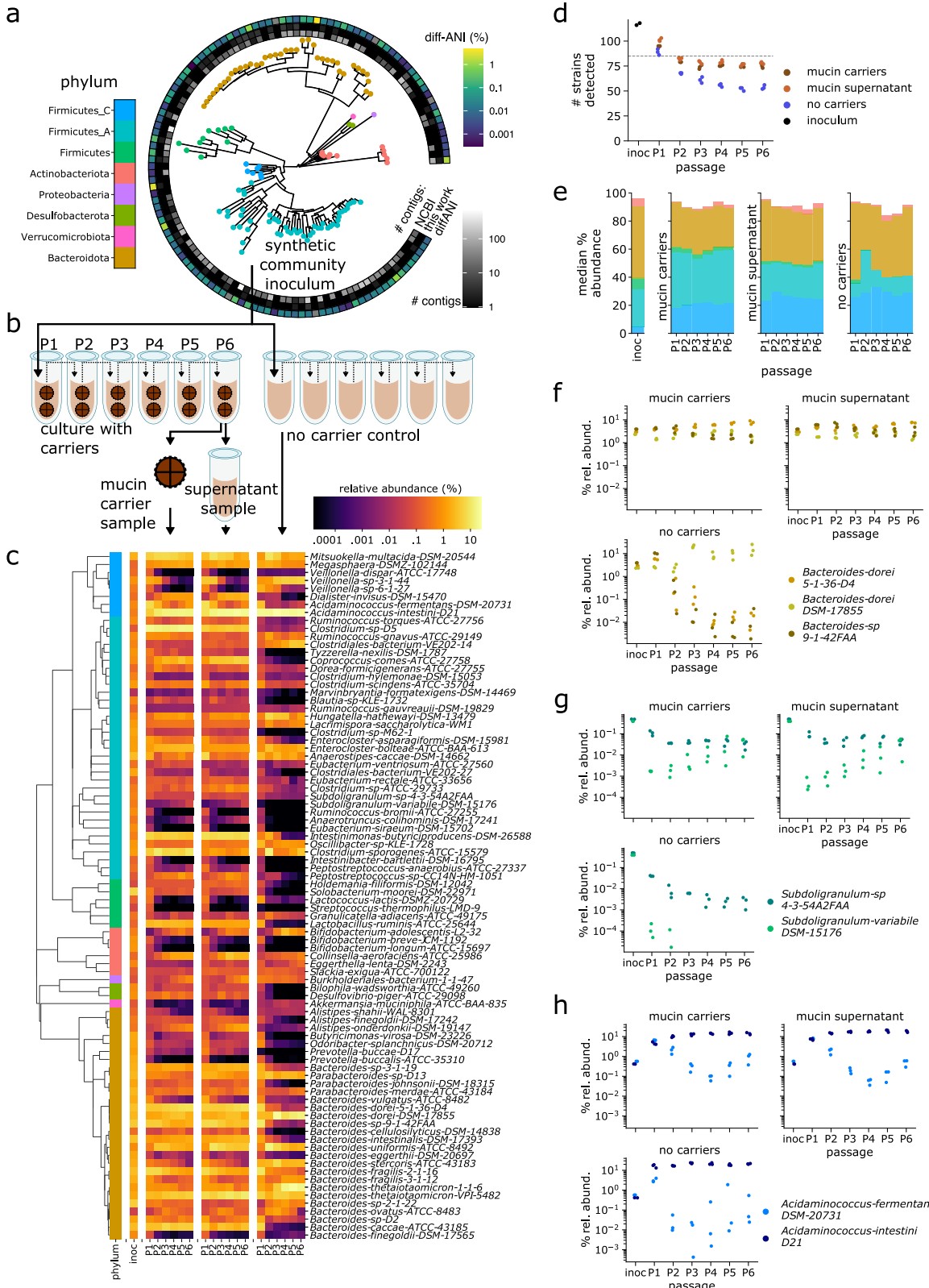

of co-existence (see Supplementary Fig. S8 for additional examples) concur with increased richness detected in carrier cultures.

## Strains exhibit distinct temporal abundance dynamics depending on culture conditions

In addition to clear differences between conditions, the frequent longitudinal sampling enabled by our in vitro approach also reveals distinct changes in community composition over time. We find that for many strains, their relative abundance drops with time (Fig. 1c), in alignment with our finding that overall community richness decreases toward later passages. However, we also observe with our mucin-carrier cultures a subset of strains that exhibit increasing abundance such as *Alistipes onderdonkii DSM-19147* and *Subdoligranulum sp. DSM-15176*, as well as strains who undergo an initial drop in abundance,

**Fig. 1 | Cultures incorporating hydrogel carriers exhibit stable, diverse communities with the co-existence of closely related strains. a** We generate high-quality genomes for each strain in a 123-member microbial community, representative of taxa in the human gut. De novo-generated genomes are more contiguous than closest previously available NCBI genomes, and represent exact matches to our strains. Six strains fail to grow, yielding a final 117-member community. **b** We use this 117-member community to inoculate cultures incorporating mucin carriers as well as non-carrier controls. We passage (P) each culture 6 times (3 days between passages), independently sampling bacterial DNA from carrier, supernatant, and no-carrier control at each timepoint for metagenomic sequencing. **c** We use NinjaMap to estimate relative abundances from metagenomic sequencing data. Here, we plot median abundance of each strain at each passage timepoint, across experimental conditions. **d** Number of detected strains after culture stabilization (-P3 and later) is higher in carrier versus no-carrier cultures,

indicating enhanced community richness. Gray dashed line indicates median number of strains detected (85) using same threshold with the 119-member hCom2 community in mice[26]. **e** Addition of carriers leads to broad taxonomic shifts in community composition relative to no-carrier control, visualized here at the phylum level. **f** Strain-resolved abundance patterns of three *B. dorei* strains (*ANI* > 99%) in our community demonstrates stable co-existence enabled by addition of carriers, compared with dominance of a single *B. dorei* strain without carriers. **g** Strain-resolved abundance patterns of two *Subdoligranulum* strains (*ANI* ~80%) in our community demonstrates stable co-existence enabled by addition of carriers. *Subdoligranulum variabile DSM-15176* in particular also exhibits increasing abundance over passage timepoints. **h** Strain-resolved abundance patterns of two *Acidaminococcus* strains (*ANI* ~80%) demonstrates more stable co-existence when cultured in the presence of carriers.

followed by recovery such as *Butyricimonas virosa DSM-23226* and *Anaerotruncus colihominis DSM-17241* (Fig. 1c and Supplementary Figs. S7 and S8). In these strains, abundances are still actively increasing even by the final passage P6 (18 days after initial inoculation), despite overall community richness stabilizing past passage 3. These temporal changes are also dependent on culture condition. For instance, while *Alistipes onderdonkii DSM-19147* increases in cultures with mucin carriers, cultures without carriers are predominated by its close relative *Alistipes shahii WAL 8301*. Conversely, *Alistipes shahii WAL 8301* decreases over time in mucin-carrier cultures. Overall, these results demonstrate that distinct temporal abundance phenotypes exist even between closely related microbial taxa, with multiple timescales defined by early and late colonizers.

## Strains exhibit distinct enrichment profiles between carrier and supernatant communities

We next characterize spatial organization *within* carrier cultures by comparing subpopulations sampled from carrier and supernatant in the same culture tube, testing our hypothesis that strain-level spatial differences occur within gut communities. First, we identify the 86 "top prevalent strains" that are abundantly observed (greater than 0.01% relative abundance) in at least 10% of passaged samples (see "Read mapping and abundance estimation"). Then, we quantify a carrier-enrichment score—defined as the log-fold change in abundance between paired carrier and supernatant samples (i.e., derived from the same culture tube)—for each strain and each passage (Fig. 2a and Supplementary Data S3). We also calculate a single aggregate, log-carrier-enrichment score for each strain using mean-over-standard deviation normalization across late passage replicate measurements (see "Hydrogel carrier-enrichment calculations" and Supplementary Data S3). These scores reflect the preference of each strain to grow attached on carriers versus in the liquid supernatant, with positive scores indicating carrier preference.

Aggregating at phylum level, we observe enrichment toward mucin carriers in Desulfobacterota, Firmicutes (primarily Bacillus-like), and Firmicutes_A (primarily Clostridia-like), and enrichment toward supernatant in Actinobacteriota, Bacteroidota, and Firmicutes_C (primarily Negativicutes-like), with no obvious time-dependent signal (Fig. 2b). At strain level, we find a diverse range of enrichment profiles over time (Fig. 2a), including several strains with opposite enrichment relative to their phylum. For instance, *Bacteroides caccae ATCC-43185* prefers carriers, while *Clostridiales bacterium VE-202-14* from phylum Firmicutes_A prefers supernatant. Moreover, closely related strains can exhibit different enrichment phenotypes: *Bacteroides dorei 5-1-36-D4* and *DSM-17855* exhibit similar abundance in supernatant and carrier (log enrichment scores ≈ 0), but *Bacteroides sp. 9-1-42FAA* displays consistent enrichment toward supernatant (log enrichment scores <0, see Fig. 2c). *Subdoligranulum variabile DSM-15176* and *Acidaminococcus fermentans DSM-20731* associate with mucin carriers more than their respective counterparts, *Subdoligranulum sp. 4-3-*

*54A2FAA* (Fig. 2d) and *Acidaminococcus intestini D21* (Fig. 2e). These findings support our hypothesis that distinct strain-level spatial organization occurs within gut communities.

As an external validation, we compare our in vitro carrier-enrichment results against an in vivo dataset[15] with paired mucosal and lumen samples (see "Comparison with in vivo dataset" and Supplementary Data S6). We find our in vitro carrier-enrichment scores are similar to in vivo mucosal-enrichment scores when comparing our strains to the corresponding species from the in vivo data (Supplementary Fig. S12 and Supplementary Data S7). We also observe general agreement at phylum level: Bacteroidota is enriched toward supernatant and lumen, while Firmicutes_A and Firmicutes are enriched toward carrier and mucosa. However, discrepancies also exist, as Actinobacteriota is enriched toward supernatant in vitro but mucosa in vivo (Supplementary Fig. S12). These results suggest that our experimental platform provides a close—though not exact—approximation of in vivo structure.

## Growth rate estimates using peak-to-trough ratio analysis

To explore if growth rate differences between carrier and supernatant conditions may be driving observed abundance enrichments, we apply the peak-to-trough ratio (PTR) method[36,37] to coverage from our metagenomic sequencing in order to estimate growth rates across strains, timepoints and culture conditions (Supplementary Fig. S11). In the strains for which read depth is sufficient to make a PTR estimate, most PTR values are close to 1, suggesting the culture mostly reaches stationary phase by the time of sampling (3 days after each passage), though there are some exceptions such as *Mitsuokella multicida DSM-20544* and *Lactobacillus ruminis ATCC-25644* which consistently exhibit higher ratios than 1, indicative of active growth. However, we observe no strains with consistent differences in PTR values *between* cultures sampled from the mucin carriers compared with corresponding supernatant. This indicates that observed mucin-carrier strain enrichment/depletion largely cannot be attributed to growth rate differences between conditions.

## Phylogenetic regression predicts genes associated with mucosal colonization

We next test for statistical associations between carrier enrichment and underlying microbial genotypes, evaluating our hypothesis that key microbial genes may regulate spatial organization in the gut. Using kofamscan[29] to comprehensively search all genomes against all defined KO families (Supplementary Data S4), we generate a genotype matrix consisting of 9857 KOs detected in the 86 top prevalent strains. Each entry in this 86 × 9857 matrix corresponds to maximum kofamscan/hmmer bitscore hit for a particular KO in a particular genome (Fig. 3a)—higher scores reflect gene presence. We then test for each of the 9857 KOs whether its genotype pattern across the 86 top prevalent strains is significantly associated with the corresponding pattern of carrier-enrichment scores (phenotype). We perform

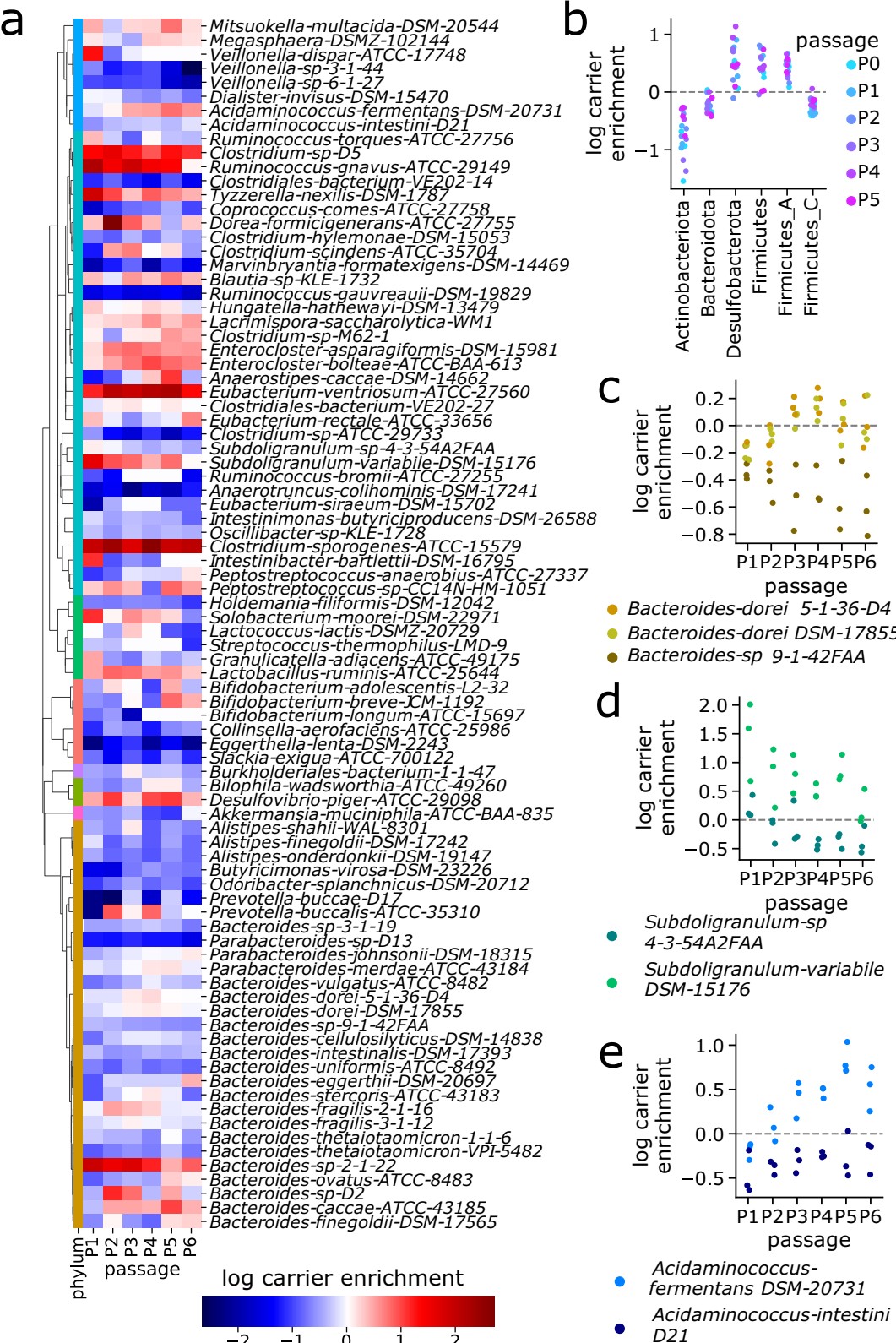

**Fig. 2 | Strain-level differences exist between mucin carrier and supernatant communities.** **a** Strains exhibit different carrier-enrichment phenotypes, both within and between clades—positive (red) scores indicate higher relative abundance on carriers versus supernatant. **b** Aggregated at phylum level, taxa exhibit evidence of distinct spatial structure: Desulfobacterota, Firmicutes and Firmicutes_A are enriched on carriers, while Actinobacteriota, Bacteroidota and Firmicutes_C are enriched in supernatant. **c** One of the three *Bacteroides dorei* strains (*sp. 9-1-42FAA*) exhibits consistent carrier depletion relative to the other two strains (*5-1-36-D4* and *DSM-17855*). **d** *Subdoligranulum variabile DSM-15176* exhibits consistent carrier-enrichment relative to the closely related strain *Subdoligranulum sp. 4-3-54A2FAA*. **e** *Acidaminococcus fermentans DSM-20731* exhibits consistent carrier enrichment relative to the closely related strain *Acidaminococcus intestini D21*. In addition, carrier enrichment of *Acidaminococcus fermentans DSM-20731* increases with time towards later passages.

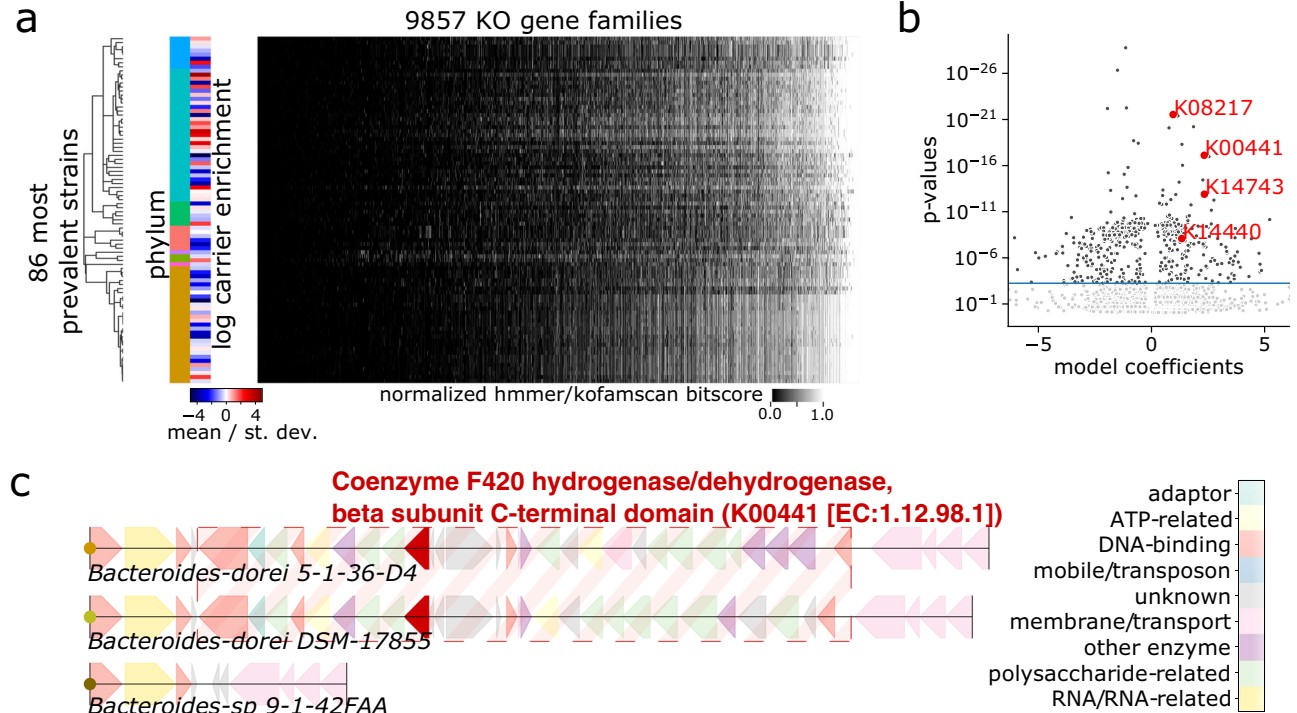

**Fig. 3 | Phylogenetic regression identifies genes associated with mucin-carrier enrichment. a** Phylogenetic regression identifies significant associations between log-carrier-enrichment score (red/blue indicates positive/negative carrier enrichment, respectively) and gene presence–absence patterns (lighter/darker shades of gray indicate gene presence/absence, respectively) across the 80 top prevalent strains detected in passaged samples. We use this model to test a total of 9857 KEGG KO gene families determined using kofamscan[29], accounting for phylogenetic relatedness between strains assuming Brownian motion along evolutionary branches. **b** Volcano plot of two-sided phylogenetic linear regression model using Brownian Motion model for covariance, where each dot represents one KEGG KO–horizontal line at FDR=0.01. Horizontal axis is clipped at 0.1 and 99.9 percentiles, highlighted gene families colored in red. **c** *Bacteroides dorei 5-1-36-D4* and *DSM-17855* both harbor a coenzyme F420 dehydrogenase gene (KEGG KO K00441) colocalized amongst LPS/EPS-related gene clusters–these features are collectively missing from the corresponding region in the *Bacteroides sp. 9-1-42FAA* genome. Plotted genome coordinates: *Bacteroides dorei 5-1-36-D4* 4619828-4663627bp, *Bacteroides dorei DSM-17855* 4750505-4793496bp, and *Bacteroides sp. 9-1-42FAA* 4767641-4780151bp.

significance tests using phylogenetic regression with phylolm[31] to account for evolutionary relationships between strains (see "Phylogenetic regression").

Our approach identifies 244 KO families significantly associated with increased enrichment on mucin carriers relative to supernatant, applying Benjamini/Hochberg false discovery rate (FDR) correction at a threshold of FDR<0.01 to account for multiple hypothesis testing (Fig. 3b, see also "Phylogenetic regression" and Supplementary Data S8). Out of these KOs, we highlight several illustrative examples whose genotype patterns align with differential carrier enrichment in the *B. dorei*, *Subdoligranulum* and *Acidaminococcus* strains featured in Fig. 2c–e. From the three *B. dorei* strains, we find two KO gene families in particular – K00441 (coenzyme F420 hydrogenase subunit beta [EC:1.12.98.1], Figs. 3c and 4a) and K08217 (MFS transporter, *DHA3* family, macrolide efflux protein, Fig. 4b)–which have strong homology hits in *Bacteroides dorei 5-1-36-D4* and *DSM-17855*, but not in *Bacteroides sp. 9-1-42FAA*. Mapping the K00441 coenzyme F420 hydrogenase hits to their genomic loci in *5-1-36-D4* and *DSM-17855*, we find the gene resides in the midst of lipo/exopolysaccharide (LPS/EPS) biosynthesis gene clusters (Fig. 3c). To address whether this coenzyme F420 hydrogenase EPS/LPS cluster colocalization is specific to *B. dorei* or appears more generally in microbial genomes, we perform gene neighborhood analysis across all 123 strain genomes to search for KOs enriched within 10 kilobases (kb) of K00441 annotated genes. We find 107 hits (see "Gene neighborhood enrichment test" and Supplementary Data S5), which are dominated by KOs with LPS/EPS biosynthesis functions, including numerous glycosyltransferase, epimerase, sugar-reductase, polysaccharide membrane transporter genes, suggesting a previously uncharacterized link

between coenzyme F420 hydrogenase and microbial LPS/EPS production.

Beside K00441 and K08217 in *B. dorei*, we also note a strong hit to a DEAD box helicase gene family – K14440, *SWI/SNF*-related matrix-associated actin-dependent regulator of chromatin subfamily A-like protein 1 [EC:3.6.4.12]–in *Subdoligranulum variabile* DSM-15176 (Fig. 4c) and a membrane protease gene family–K14743, membrane-anchored mycosin *MYCP* [EC:3.4.21.-]–in *Acidaminococcus fermentans DSM-20731* (Fig. 4d) which are absent in their less carrier-enriched relatives. Intriguingly, LPS/EPS biosynthesis[38–42], membrane transporters/efflux pumps[43–49], membrane proteases[50–55], and DEAD box helicase gene regulators[40,56–59] all have known links to biofilm formation and adhesion. Aggregating all 244 carrier-associated KOs by KEGG BRITE gene categories, we identify several BRITE categories enriched for significant KOs, representing antibiotic resistance genes, glycosyltransferases (E.C. 2.4), phosphotransferases (E.C. 2.7), transcriptional regulators, and proteases (Supplementary Data S11), further supporting the importance of these gene functions in surface/mucosal colonization.

Testing for clade-specific effects using within-phylum phylogenetic regression, we find K14440 and K14743 to be among the most significant hits in Firmicutes/Firmicutes_A/Firmicutes_C, while K00441, K14743 and K08217 are among the most significant hits for Bacteroidota (Supplementary Data S9). As an external validation, we repeat our workflow using the Suez et al. in vivo dataset[15] to identify a list of KOs associated with mucosal enrichment (Supplementary Data S10), and find statistically significant overlap between genes associated with carrier enrichment in vitro and genes associated with mucosal enrichment in vivo, (*log − odds*

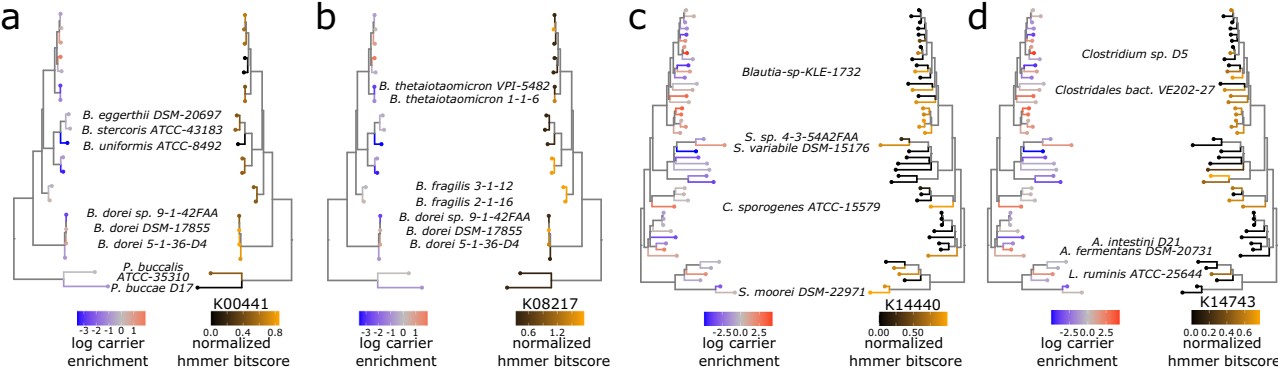

**Fig. 4 | Phylolm-identified gene families have presence patterns that align with differential carrier enrichment. a** Comparison of carrier-enrichment pattern (left) with gene presence pattern (right) of K00441 coenzyme F420 hydrogenase subunit beta [EC:1.12.98.1], across family Bacteroidaceae strains. **b** Comparison of carrier-enrichment pattern (left) with gene presence pattern (right) of K08217 MFS transporter, *DHA3* family, macrolide efflux protein, across family Bacteroidaceae strains. **c** Comparison of carrier-enrichment pattern (left) with gene presence pattern (right) of K14743 membrane-anchored mycosin *MYCP* [EC:3.4.21.-], across phylum Firmicutes_A, Firmicutes_C, and Firmicutes strains. **d** Comparison of carrier-enrichment pattern (left) with gene presence pattern (right) of K14440 *SWI/SNF*-related matrix-associated actin-dependent regulator of chromatin subfamily A-like protein 1 [EC:5.6.2.-], across phylum Firmicutes_A, Firmicutes_C, and Firmicutes strains.

*ratio* = 3.99, 95% *CI* 2.89–5.65, *P* < 2.2 × 10⁻¹⁶, two-sided Fisher's exact test, Supplementary Fig. S13). Thus, we confirm that measurements from our in vitro synthetic community cultures are sufficiently detailed to inform a computational gene-level analysis of gut spatial organization, revealing that genes related to biofilm formation and adhesion likely play key roles in modulating the physical structure of gut microbial communities.

### Strain enrichment on carriers is associated with the presence of lipo/exopolysaccharide biosynthesis gene clusters

To explore mechanisms of community structure beyond individual genes, we next investigate carrier enrichment of biosynthetic gene clusters (BGCs). We use deepBGC[60] to search for BGCs across our strain genomes, annotate BGCs based on their KEGG KO presence, and apply hierarchical clustering to categorize 1103 detected BGCs into 256 groups with similar KO co-occurrence patterns (Fig. 5a and Supplementary Data S12). We then map the presence/absence of each of these 256 BGC groups against the 86 top prevalent strains in our experiment (Supplementary Fig. S14), and apply phylogenetic regression to test for associations between carrier enrichment and BGC groups.

Our approach yields a total of 7/256 significant BGC groups positively associated with carrier enrichment (Fig. 5b), the three largest of which consist of 18 or more BGC representatives (BGC-group 157 – see Fig. 5c, BGC-group 120, and BGC-group 69). Filtering for the most common KEGG KOs in each of these BGC groups, we discover that BGC-group 157 and BGC-group 120 consist of likely EPS-related gene clusters, typified by glycosyltransferase, epimerase and other EPS-related KOs (Fig. 5d and Supplementary Data S13). BGC-group 69 consists largely of gene clusters populated by membrane transporter genes. KOs in other carrier-enriched BGC groups include more polysaccharide-related genes (BGC groups 198, 186, 161) and AraC transcriptional regulator genes (BGC-group 34). These findings at the BGC level further reinforce our KO-level results, showing that membrane-related functions such as LPS/EPS and transporters, as well as key gene regulators, likely contribute to spatial organization in our in vitro model of the human gut.

### Reductionist comparison with plain-agar carriers reveals strain-specific effects of mucin

Our in vitro approach allows precise control over the culture environment, enabling a reductionist comparison to disentangle the specific contributions of adding a hydrogel surface from the effects of the mucin. To explore this, we grow our inoculum in cultures with 1% agar carriers (plain-agar, i.e., no mucin). These comparisons allow us to distinctly measure the differences that arise from (i) introducing a hydrogel surface, and (ii) introducing mucin into the hydrogel. Using PCA visualization (Supplementary Fig. S9), we demonstrate that cultures from the five different culture conditions (liquid-only, mucin-agar carrier/supernatant, and plain-agar carrier/supernatant) all exhibit distinct community profiles (Supplementary Fig. S7).

We observe that similar to mucin-agar, plain-agar carrier culture also exhibits overall enhanced richness compared with liquid-only culture (Supplementary Fig. S6). This suggests that increased richness results largely—but not entirely—from having a physical surface to colonize rather than the nutrients provided by mucin. These results are reminiscent of similar effects in bacterial biofilms, where increased diversity has been attributed to expanded spatial niches and reduced competition[61–63]. Moreover, by comparing carrier/supernatant enrichment scores between mucin-agar and plain-agar conditions (Supplementary Fig. S10), we find overall correlation across strains as expected (enrichment on plain-agar carriers is a good predictor of enrichment on mucin-agar carriers), but with several notable exceptions, such as *Akkermansia muciniphila ATCC-BAA-835*, *Eubacterium ventriosum ATCC-27560* and *Bacteroides caccae ATCC-43185*, which exhibit higher than expected mucin-agar enrichment compared to plain-agar. At phylum level, we find that Desulfobacterota is enriched on mucin-agar carriers, but not enriched on plain-agar carriers. Similar patterns are observed when quantifying mucin-agar-carrier/plain-agar-carrier-enrichment scores (Supplementary Data S3). Applying the phylogenetic regression approach discussed earlier, we use these mucin-agar-carrier/plain-agar-carrier-enrichment scores to search for and categorize genes associated with elevated mucin-specific abundance, identifying phosphotransferases (E.C. 2.7) and oxidoreductases (E.C. 1.2) among mucin enriched gene categories (Supplementary Data S14 and S15). Thus, by using an in vitro approach, we are able to disentangle the effects of adding a hydrogel surface versus adding mucin and identify key taxa and genes.

## Discussion

Applying in vitro mucin-carrier culture with our defined community of human gut strains, we present here the first strain-resolved measurements of spatial structure within the context of a complex gut microbial community. We a priori generate a database of high-quality reference genomes, generating 118 new genomes, 81 of which are fully closed—this approach enables high taxonomic-resolution abundance measurements using metagenomic sequencing, while recapitulating

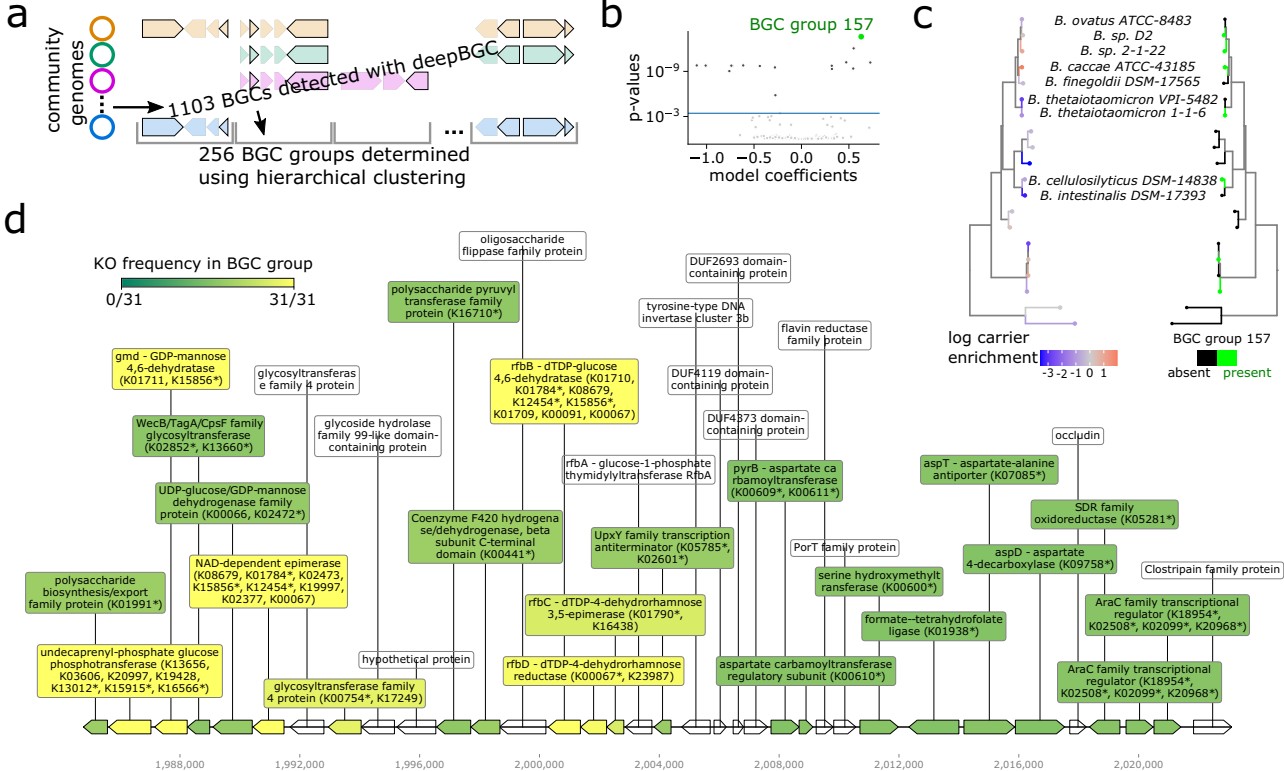

**Fig. 5 | Strain enrichment on mucin carriers is associated with exopoly-saccharide gene clusters. a** Schematic of approach used to generate and group BGCs across strains using deepBGC and hierarchical clustering. **b** Volcano plot of phylogenetic linear regression model using Brownian Motion model for covariance, each dot represents one BGC-group, horizontal line at FDR = 0.01 cutoff. Top hit BGC-group 157 highlighted in green. **c** Comparison of carrier-enrichment

pattern (left) with presence pattern (right) of BGC-group 157 across strains in family Bacteroidaceae. **d** Example of a representative gene cluster in BGC-group 157 from the *Bacteroides* strain with the highest carrier-enrichment score (*Bacteroides-sp.2-1-22_cluster_1_1984776-2023109.1*). Gene label colors reflect frequency of KO family among all BGCs in group—in cases where a gene maps to multiple KOs, * marks mapped KO with highest frequency in BGC group.

aspects of spatial structure and taxonomic complexity in the gut microbiome. These measurements show with high taxonomic resolution how a complex gut microbial community is spatially organized upon introduction of carriers, demonstrating that carriers enhance community richness to a level similar to in vivo observations, including instances of co-existence between closely related strains. We find clear enrichment signals *within* carrier cultures where certain strains prefer to grow on the carriers versus in the supernatant, or vice versa. Carrier-enrichment phenotypes can differ significantly even between closely related strains, supporting our hypothesis that spatial organization in the gut occurs at strain level and trends would be missed at coarser taxonomic resolution.

While an in vitro approach can never fully realistically recapitulate all aspects of the in vivo gut environment, our choice of in vitro culture offers two key capabilities that are less feasible in vivo. First, we are able to distinctly sample hydrogel-attached and liquid phase culture fractions with frequent longitudinal measurements. These measurements reveal the existence of distinct temporal abundance phenotypes exist even between closely related microbial taxa, with multiple timescales defined by early and late colonizers. Second, we are able to control the culture environment in a tunable manner, in particular with respect to the hydrogel surface. Our observation that the addition of mucin to the hydrogels (compared with plain-agar hydrogel carriers) shifts carrier-enrichment profiles across the community aligns with earlier work demonstrating that mucin is capable of modulating biofilm formation in bacteria[64].

Combined with high-quality reference genomes, our shotgun metagenomic measurements enable estimates of relative growth rates using the PTR method[36,37]. We do not observe consistent differences in

PTR scores between cultures sampled from the mucin carriers compared with the surrounding liquid phase, suggesting that observed mucin-carrier strain enrichment/depletion cannot be attributed overall to growth rate differences between conditions. We therefore hypothesize that differences in attachment to the hydrogel surface may also play a key role.

An additional benefit of strain-resolved shotgun metagenomics is that we can identify gene families that specifically occur in strains with carrier-enrichment (or depletion) phenotypes. We do so using phylogenetic regression, a rigorous statistical approach that adjusts for evolutionary relationships between strains. This analysis identifies several gene families related to microbial adhesion and biofilm formation, including efflux pumps (e.g., K08217) that are known to mediate collective biofilm phenotypes such as quorum sensing and antibiotic resistance[43–49], and membrane proteases (e.g., K14743) which can enhance motility / colonization on surfaces[50–55]. We also find genes involved in biosynthesis of LPS/EPS which are known to mediate bacterial adhesion[38–42], such as glycosyltransferase and epimerase genes, as well as a particular gene family K00441 (coenzyme F420 hydrogenase subunit beta [EC:1.12.98.1]) for which we report significant genomic colocalization with other known LPS/EPS genes, suggesting a previously uncharacterized functional link. We also find several groups of biosynthetic gene clusters containing membrane transporters and LPS/EPS genes associated with carrier enrichment. Beyond membrane-associated functions, our analysis also highlights regulatory genes such as *SWI/SNF* DEAD box helicases (K14440). Intriguingly, such genes have not only been shown to be involved in biofilm formation[40,56–59], but also specifically drive expression of efflux pumps and LPS/EPS genes[56]. We speculate that in mucosa-associated

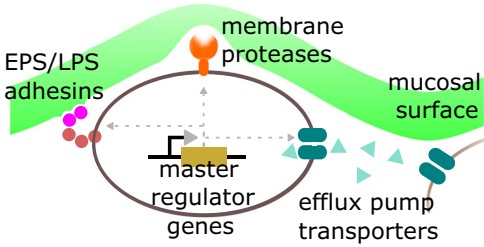

**Fig. 6 | Schema for how mucosal-associated genes may regulate spatial structure.** Depiction of proposed framework where in mucosa-associated taxa, regulatory genes serve as master switches for microbial functions that increase mucosal colonization fitness such as LPS/EPS, membrane transporters/efflux pumps, and proteases.

taxa, key regulator genes act as master switches for a host of bacterial functions that alter outer membrane composition to enhance biophysical interactions with the mucosal surface and thus increase mucosal colonization fitness, leading to global spatial organization of these taxa towards the mucosa (Fig. 6).

We conclude by noting several limitations to our work and point to areas for further exploration. First, as mentioned above, our in vitro approach is unable to recapitulate all aspects of the in vivo gut environment, such as oxygenation gradients present at the luminal boundary, or effects of the host immune system. While existing traditional fecal sampling techniques do not attempt to resolve mucosal versus lumenal bacteria, recent findings that fecal pellets may retain mucosal/lumen structure found in the gut[65] suggest a direction for future work that uses more careful sampling to obtain in vivo spatially resolved, longitudinal measurements of gut communities. Second, our gene-level analysis only shows statistical associations—not causal mechanisms—between genotypes and carrier enrichment, meaning hits should be cautiously interpreted as potential genetic factors deserving of follow-up investigation. Synthetic biology in genetically tractable gut strains can be used to test our predictions by altering the expression of identified gene families using gene knockout, knockdown or knockin experiments[66–68]. Third, while our in vitro results generally parallel those from earlier in vivo work[15,26], we do find a few discrepancies (e.g., carrier depletion of Actinobacteriota), meaning our current platform provides a close but still imperfect replica of the in vivo gut environment. More realistic culture conditions can be explored, potentially through modification of media conditions (e.g., the addition of bile acids, different carbon sources). Fourth, our current approach based on metagenomic sequencing provides accurate quantification of strain and gene abundance, but it does not assay gene expression or spatial localization on carriers. Future work using gut microbial metatranscriptomic analysis[69,70] and multiplexed FISH imaging[71–73] can greatly complement current capabilities and mitigate these shortcomings. Fifth, it remains unclear how strain-strain interactions affect structure. Follow-on studies with our platform that incorporate strain dropout can address these questions. Finally, in addition to strains from healthy Western guts, future work should incorporate taxa found in dysbiotic and non-Western guts to explore how spatial structure varies between healthy and diseased states, and across global populations. Ultimately we believe the platform presented here has the potential to transform the standard for in vitro investigation of gut microbiota, in a manner that recognizes the important interplay between spatial structure and strain-level ecology.

## Methods

### Hybrid assembly of microbial isolates
Strains are cultured in isolation until stationary phase, followed by DNA extraction using phenol chloroform. DNA is sequenced using both Oxford Nanopore long-read and Illumina short-read sequencing, followed by hybrid assembly using custom bioinformatic workflow (Supplementary Fig. S1) built using Unicycler[74], LRScaf[75], and TGS-GapCloser[76]—workflow is available as docker images, see Software availability below.

### Community phylogeny
Phylogenetic tree structure of the community is generated using GTDB-tk[77], using our genome assemblies as input.

### Genome annotation and gene classification
Genomes are annotated using NCBI PGAP[78]. Predicted protein sequences are then mapped using kofamscan[29] to the to KEGG Orthology database.

### Mucin-carrier preparation
Mucin carriers—also previously referred to as microcosms—are prepared based on described protocols[27,79] using a solution of 0.5% porcine mucin (Sigma M2378) and 1% agar (BD 214030). Mucin-agar solution is autoclaved and poured over K1 biofilm carriers (Evolution Aqua MEDIAK1), and then allowed to solidify. Mucin-agar embedded carriers are then extracted using tweezers. Mucin free agar-only carriers are prepared using 1% agar solution.

### In vitro culture of synthetic community with mucin carriers
To construct the full in vitro synthetic community, we first culture each strain in isolation in 1.8 mL of its preferred media in a 96-well deep-well plate (Supplementary Data S1). Because of the large range of growth rates and stationary phase cell densities, strains are inoculated in a staggered fashion with slow growers inoculated 3 days prior and fast growers inoculated 1 day before community assembly. Fastidious growers are cultured in 10 mL and concentrated to increase final cell density. Individual isolate cultures are sequenced to verify purity. On the day of community assembly, cell density for each strain is estimated using OD measured on a plate reader (BioTek Epoch). Using this measurement, each strain is normalized to a maximum OD of 0.3 using liquid handling robotics. Cultures are pelleted and washed with PBS, and then combined to form a mixture of 117 strains (epMotion 5073). Strains are combined in an anaerobic environment equipped with automated liquid handling in order to reduce potential cross contamination and other human errors when concurrently handling many strains (Supplementary Fig. S3).

Following assembly of our bacterial community, the mixture is used to inoculate cultures in 15-mL tubes comprising MEGA media[34] and 5 carriers each. Cultures are left to grow at 37 °C in anaerobic conditions for 3 days without agitation, at which point they are passaged. Passaging consists of transferring a single carrier from the old culture tube to a new culture tube. This process is repeated five times for a total of six passages. For each subsequent passage, the previously transferred-in carrier is discarded prior to transferring of a carrier to the next culture—due to the shape of the tubes and size of the carriers, the carriers are unable to move around other, which means the vertical stacking order of the carriers does not change and the previously transferred carrier remains at the top. After cultures are grown for three days, this top carrier (which came from the previous passage) is discarded, the next carrier underneath is transferred to the subsequent culture passage, while three of the remaining carriers are harvested for DNA isolation, with one backup carrier. For liquid-only cultures, inoculating loops (Fisherbrand 01-189-165) are used for passaging. Supernatant and carrier samples are saved and frozen at each passage point, prior to DNA isolation.

For each condition, we culture the community in biological triplicate cultures (i.e., three separate culture tubes). Each culture tube is sampled with technical triplicates—for carrier samples, we pick three carriers out of each culture tube to store at −80 °C prior to DNA

extraction, while for supernatant and liquid-only cultures, we take three separate 1-mL aliquots from each tube, pellet, then store at −80 °C prior to DNA extraction. This yields a total of nine read libraries for each passage and experimental condition. The initial inoculum communities are sampled in duplicate, each sample sequenced three times each. Experiments were carried out at CZ Biohub Lokey Facility at Stanford University.

## DNA extraction, library prep, and sequencing
DNA is extracted from supernatant and carriers using ZymoBIOMICS 96 DNA Kit and bead beating with 0.1-mm glass beads (Benchmark Scientific D1031-01). Extracted DNA from each sample is quantified in 384-well plates on a fluorescent plate reader (BioTek Neo2) using the Quant-iT PicoGreen assay (ThermoFisher). To generate input DNA for our high-throughput and low-volume Nextera XT library preparation process, DNA samples are normalized to at maximum of 0.2 ng/μL in a 384-well plate using a low-volume cherry-picking liquid handler (SPT). Library preparation is done in 384-well plates using a low-volume 16-channel liquid handler (SPT) and follows the chemistry of the Nextera XT process but in a total volume of 4 μL in order to reduce library preparation cost. Libraries are quantified again using the Quant-iT PicoGreen assay and normalized. After pooling and cleaning using Ampure XP beads (Beckman), libraries are sequenced on a Novaseq 6000 (Illumina) to a mean depth of $1.2 \times 10^7$ read pairs per sample. In addition to DNA derived from microbial communities, we also sequenced all input strains used to construct the community to ensure strain purity and identity.

## Read mapping and abundance estimation
Read mapping is performed with NinjaMap as previously described[26], using our de novo-generated genomes as reference database. Briefly, reads are aligned to genome sequences, with only perfect unique matches considered in the first round. Ambiguous reads are held in escrow for the first round, and subsequently assigned in a statistically weighted manner determined by initial abundance estimates from the first round of alignment. This generates relative abundance and horizontal genome coverage estimates for each strain in each sample's read library. We consider a strain present in a sample if it exceeds a 1% horizontal coverage and 0.0001% relative abundance cutoff. Out of all 270 passage samples (6 passages × 5 experimental conditions−mucin-agar carriers, mucin-agar supernatant, plain-agar carriers, plain-agar supernatant, no-carriers− × 9 replicates), we use a stricter prevalence cutoff of 10% presence at 0.01% abundance (i.e., present in 27 or more samples) to focus on the top prevalent strains (total of 86 strains). For downstream abundance-related analysis, we collapse technical (i.e., within tube) triplicates to their median abundance measurements while considering biological (different culture tubes) triplicate measurements separately. Supplementary Data S2 lists relative abundance and horizontal coverage across strains, passages, replicates and experimental conditions.

## Hydrogel carrier-enrichment calculations
For each strain, and passage, carrier-enrichment scores are calculated as log ratio of carrier to supernatant abundance, for three biological replicates, replacing zeros with half-minimum nonzero value prior to log-transform. For each strain, a single aggregate enrichment score is generated by taking mean over standard deviation of 12 log ratio scores in the late passages (P3-6, 4 passages × 3 biological replicates). Supplementary Data S3 lists enrichment scores per strain.

## Gene neighborhood enrichment test
Based on results from kofamscan for each gene in each genome, a gene is annotated with a KO-label if it exhibits overlap greater than 0.5 × coverage with the KO's pHMM model, as well as a bitscore greater

than 0.5 × the KO's bitscore threshold. We count the frequency of all annotated KOs within 10 kb of K00441-labeled genes across the full community genome database. To generate $P$ value estimate of this measured frequency, we compare it against a null distribution generated by 1000 random gene order permutations. In each of these 1000 permutations, we randomly reassign gene labels within each of the 123 genomes prior to conducting frequency counts. $P < 0.01$ indicates 990 or more times out of 1000, the actual co-occurence of a particular KO within 10 kb of K00441 is greater than random.

## Phylogenetic regression
For each KO family, and each strain, we determine the maximum hmmer bitscore hit to the KO's pHMM out of all the strain's proteins. Aggregating across KOs and strains, this yields a strain-by-KO genotype matrix, where each entry is the highest bitscore value−higher bitscores indicate gene presence. We then test for association between this genotype and carrier-enrichment score (phenotype). While such genotype-phenotype tests are in many ways similar to those conducted in genome association studies (GWAS), the application of ordinary least squares (OLS) regression, often used in GWAS, is not appropriate here due to phylogenetic relationships between strains. These relationships mean that assumptions of independence between measurements inherent to OLS are violated. We confirm the presence of a non-star phylogeny between strains by generating a phylogenetic tree based on strain genomes, using bac120 multiple sequence alignment with GTDB-tk[77] (Fig. 1a). Therefore, to account for this phylogenetic relatedness, for each KO we apply phylogenetic regression to test for significant association between mucosal-enrichment scores and maximum hmmer bitscore (standard scaled) across strains. We implement this test using the R package phylolm[31], assuming a Brownian motion model along evolutionary branches, using the bac120 phylogenetic tree as input. This generates effect size estimates and $P$ values for every KO. We filter KOs for significance with Benjamini-Hochberg false discovery rate (FDR) correction to account for multiple hypothesis testing, applying a FDR<0.01 cutoff. In addition to running phylogenetic regression across all 86 top prevalent strains, we also run these models across subsets of these strains grouped by phylum to search for clade-specific hits. For this analysis, we group Firmicutes, Firmicutes_A, Firmicutes_C phyla into a single clade.

## Comparison with in vivo dataset
We analyze from the Suez et al.[15] dataset all read libraries from untreated (i.e., naive) individuals, for whom lumen and mucosal reads were available from cecum, descending colon, and terminal ileum, i.e., a total of 6 read libraries per individual. We first use Kneaddata (part of the Biobakery suite[80]) to perform host (i.e., human) filtering of read sequences, resulting in 13 individuals for which all 6 libraries exceed a read depth of $10^4$ reads (Supplementary Data S6). For these 13 individuals, we obtain abundance estimates at all 6 sites by mapping reads to UHGG database using Kraken2[2,35]. We then calculate normalized mucosal-enrichment scores for each species defined as log ratio of mucosal to lumen abundance. Score are normalized by taking mean-over-standard deviation across all individuals and sites (13 individuals × 3 sites−cecum, descending colon and terminal ileum−for 39 total measurements). We determine gene presence−absence for these species, across KOs, by using kofamscan to search the UHGG pangenome database[2], and then apply phylogenetic regression as described above to test for associations between gene presence−absence and mucosal-enrichment score across UHGG species. The regression uses enrichment scores from 676 species detected with greater than 0.01% relative abundance in at least 10% of in vivo read libraries, which contain a total of 12,822 detected KEGG KO gene families (Supplementary Data S7 and S10).

## Extraction and grouping of biosynthetic gene clusters using DeepBGC and hierarchical clustering

DeepBGC[60] is used to extract BGCs from our de novo genomes. For each identified BGC, we generate a list of present KOs based on if contained genes map to KO's pHMM with overlap greater than $0.5 \times$ coverage, as well as a bitscore greater than $0.5 \times$ the KO's bitscore threshold. We filter out BGCs with fewer than 3 present KOs, and then use hierarchical clustering to cluster all remaining BGCs based on their binary KO presence/absence profile into 256 BGC groups, applying a Jaccard distance metric. We then map presence of each BGC group within community strains, and use this presence/absence matrix to test for associations with carrier enrichment applying phylogenetic regression as described above.

## Peak-to-trough growth rate estimation

Peak-to-trough growth estimates are done with the iRep[37] software package, using the bPTR implementation to take advantage of the availability of closed genomes.

## Data visualization

Custom python and R scripts were developed for data visualization, using the following packages: scipy[81], pandas[82], numpy[83], seaborn[84], python[85], ipython[86], jupyter notebook[87], statsmodels[88], dna-features-viewer[89], biopython[90], reportlab, matplotlib[91], R[92], ggtree[93], treeio[94], ggnewscale[95], phytools[96], RColorBrewer[97], tidyr[98], dplyr[99], stringr[100], ggplot2[101], cowplot[102], ape[103].

## Reporting summary

Further information on research design is available in the Nature Portfolio Reporting Summary linked to this article.

## Data availability

The in vitro culture sequencing data generated in this study have been deposited in the NCBI database under BioProject accession code PRJNA885585, see Supplementary Data S16 for additional metadata. Hybrid-assembled genomes and underlying sequencing data used to generate these genomes are deposited to BioProject PRJNA885826 and to BioProject PRJNA746600. Details for each genome are listed in Supplementary Data S1 including BioProject accession number as well as individual BioSample IDs. Key processed data generated in this study are provided in the Supplementary Data, with full data deposited in Figshare dataset gut-community-microcosms [https://figshare.com/articles/dataset/gut-community-microcosms/21094717]. GTDB, UHGG, and KEGG KO databases are publicly available.

## Code availability

Code used for analysis and visualization is available at https://github.com/xiaofanjin/gut-community-microcosms[104] Code used for hybrid assembly is available at https://github.com/FischbachLab/nf-hybridassembly[105].

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

## Acknowledgements

We thank A. Lind, P. Bradley, A. Bustion, and I.H. Riedel-Kruse for helpful discussions regarding data analysis and feedback on the manuscript; A. Cheng and M. Fischbach for sharing bacterial strains, and S. Jain for helpful discussions regarding metagenomic data analysis. We also thank the CZB Genomics Platform for support and encouragement, and are especially grateful to N. Neff, M. Tan, A. Detweiler, S. Paul, and H. Mekonen for their sequencing efforts on this project. This work is supported by funding from Chan Zuckerberg Biohub, Gladstone Institutes, NSF grant #1563159, and NHLBI grant #HL160862.

## Author contributions

X.J., F.B.Y., and K.S.P. contributed to the design and implementation of the research, to the analysis of the results and to the writing of the manuscript. V.D., X.M., J.Y., and A.M.W. contributed to the implementation of the research.

## Competing interests

The authors declare no competing interests.
