## [Peer Review File · Nature Communications]

REVIEWERS' COMMENTS

Reviewer #4 (Remarks to the Author):

In this manuscript, Jin and colleagues use a complex community of 117 human strains, and in vitro culture incorporating hydrogel carriers to study the effect of the availability of a surface for bacterial attachment on bacterial community structure. These measurements are intended to simulate and discover determinants of spatial organization in the gut. I have read the manuscript and the comments provided by previous reviewers. In my opinion, the experimental system introduced in this work is quite elegant. The scope of experiments is impressive, in particular, the authors obtained closed and near-closed genomes for all the strains in the community which enabled detailed strain-level analyses and identification of gene families associated with carrier enrichment. I have a few suggestions for further improvement of the presentation of the data.

First, the title of the paper is overstated. The culture model does not “reveal the ecology and genetics of gut microbial organization”. The paper does show the importance of the availability of a hard surface (and mucin) to maintain complexity of bacterial communities in culture. Surface adhesion is surely a factor in microbial organization in the gut but does not explain the full complexity of the ecology and genetics of the gut microbiome. This is too much of a leap. Synthetic bacterial communities are being considered as Life Biotherapeutic Products for the treatment of various diseases and the present study can provide insights into the maintenance of the complexity of these products and can provide new avenues for engineering.

Second, in response to reviewer comments, the authors assessed the effect of plain-agar carriers in addition to mucin carriers. The data indicate that the increased richness of the community after several passages is mostly due to the availability of a physical surface rather than nutrients provided by mucin. This is an important finding that needs to be included in the main text, not just the supplement. The effect of mucin in the hydrogels can be further investigated, by examining genes enriched in mucin hydrogels relative to plain agar gels. For example, the analyses of figures 3-5 can be extended to assess the effects of the surface and mucin-carriers separately.

Last, in figure 4, the authors use the term “microcosm enrichment”. This is confusing and should be removed.

Reviewer #5 (Remarks to the Author):

The responses related to the previous comments of reviewer #3 are satisfactory and have reasonably addressed the concerns raised.

We would like to thank reviewers for their time and thoughtful comments on how to improve the manuscript. Please find below a detailed response to reviewer comments. We carried out the suggested analysis and modifications, and we believe we were able to address all the reviewers' comments in full.

Reviewer #4 (Remarks to the Author):

In this manuscript, Jin and colleagues use a complex community of 117 human strains, and in vitro culture incorporating hydrogel carriers to study the effect of the availability of a surface for bacterial attachment on bacterial community structure. These measurements are intended to simulate and discover determinants of spatial organization in the gut. I have read the manuscript and the comments provided by previous reviewers. In my opinion, the experimental system introduced in this work is quite elegant. The scope of experiments is impressive, in particular, the authors obtained closed and near-closed genomes for all the strains in the community which enabled detailed strain-level analyses and identification of gene families associated with carrier enrichment. I have a few suggestions for further improvement of the presentation of the data.

- *Thank you for taking the time to read the manuscript and provide these constructive comments.*

First, the title of the paper is overstated. The culture model does not “reveal the ecology and genetics of gut microbial organization”. The paper does show the importance of the availability of a hard surface (and mucin) to maintain complexity of bacterial communities in culture. Surface adhesion is surely a factor in microbial organization in the gut but does not explain the full complexity of the ecology and genetics of the gut microbiome. This is too much of a leap. Synthetic bacterial communities are being considered as Life Biotherapeutic Products for the treatment of various diseases and the present study can provide insights into the maintenance of the complexity of these products and can provide new avenues for engineering.

- *We have updated the title to clarify that we are not presenting a comprehensive description of ecology and genetics in the gut microbiome. The new title is “Hydrogel carrier cultures of a complex gut microbial community reveal strains and genes associated with spatial organization”.*

Second, in response to reviewer comments, the authors assessed the effect of plain-agar carriers in addition to mucin carriers. The data indicate that the increased richness of the community after several passages is mostly due to the availability of a physical surface rather than nutrients provided by mucin. This is an important finding that needs to be included in the main text, not just the supplement. The effect of mucin in the hydrogels can be further investigated, by examining genes enriched in mucin hydrogels relative to plain agar gels. For example, the analyses of figures 3-5 can be extended to assess the effects of the surface and mucin-carriers separately.

- *Thank you for these suggestions. We have updated the main text to highlight the importance of providing a surface to community richness - see introduction: “We compare these results with no-carrier (i.e., liquid media only) cultures and cultures with plain-agar (i.e. no mucin) carriers to disentangle the specific effects of added mucin versus hydrogel surface, revealing that the presence of carriers serves to increase community richness, as well as mucin-specific effects.” We have also extended our analysis to look for mucin-specific effects by testing for genes associated with greater strain abundance on mucin-agar vs. plain-agar carriers. These results are now included in a dedicated Results section (“Reductionist comparison with plain-agar carriers reveals strain-specific effects of mucin”) and new Supplementary Tables S14 and S15.*

Last, in figure 4, the authors use the term “microcosm enrichment”. This is confusing and should be removed.

- *Thank you for catching this oversight. We have updated the figure to “carrier enrichment”.*

Reviewer #5 (Remarks to the Author):

The responses related to the previous comments of reviewer #3 are satisfactory and have reasonably addressed the concerns raised.

- *Thank you for taking the time to read the manuscript.*